# Study Effect of nAg Particle Size on the Properties and Antibacterial Characteristics of Polysulfone Membranes

**DOI:** 10.3390/nano12030388

**Published:** 2022-01-25

**Authors:** Gunawan Setia Prihandana, Tutik Sriani, Aisyah Dewi Muthi’ah, Affiani Machmudah, Muslim Mahardika, Norihisa Miki

**Affiliations:** 1Department of Industrial Engineering, Faculty of Advanced Technology and Multidiscipline, Universitas Airlangga, Jl. Dr. Ir. H. Soekarno, Surabaya 60115, Indonesia; aisyahdewimuthiah@gmail.com (A.D.M.); affiani.machmudah@gmail.com (A.M.); 2Department of Research and Development, PT. Global Meditek Utama, Sardonoharjo, Ngaglik, Sleman, Yogyakarta 55581, Indonesia; tsriani@iitoya.com; 3Department of Mechanical and Industrial Engineering, Faculty of Engineering, Universitas Gadjah Mada, Jalan Grafika No. 2, Yogyakarta 55281, Indonesia; muslim_mahardika@ugm.ac.id; 4Department of Mechanical Engineering, Keio University, 3-14-1 Hiyoshi, Kohoku-ku, Yokohama 223-8522, Japan; miki@mech.keio.ac.jp

**Keywords:** polysulfone membrane, ultrafiltration, silver nanoparticles, environmentally sound technologies

## Abstract

Polysulfone ultrafiltration membranes were fabricated using various sizes (20, 40, and 90–210 nm) of silver nanoparticles (nAg) blended in a dope solution. To characterize the performance and properties of the prepared membranes, scanning electron microscopy (SEM), water contact angle, protein separation, water flux, and antibacterial tests were conducted. The characterization results revealed that when nAg particles (20 nm) were blended into the base polymer PSF, the PSF/nAg blended membrane had the lowest contact angle (58.5°) and surface energy (110.7 mN/m). When experimenting with ultrafiltration using protein solutions, bare PSF and PSF/nAg-20 blended membranes gave similar values of protein rejection: 93% of bovine serum albumin (BSA) and 70% of lysozyme rejection. Furthermore, SEM studies showed that the surface pore size was reduced by adding 20 nm nAg particles in the casting solution. Most importantly, the introduction of 40 nm nAg particles reduced the growth of bacterial colonies on the membrane surface by up to 72%. These findings revealed that nAg particles are expected to be a potential modifier for the fabrication of an ultrafiltration membrane.

## 1. Introduction

Polymer membranes have been widely used as a key component in separation technologies to remove macromolecules, bacteria, viruses, and salt from any type of feed liquid in the past two decades [1,2,3]. Polysulfone (PSF) is one of the most popular polymeric materials used, among others, such as polyethersulfone, polypropylene, poly(arylene ether nitrile ketone), polyvinylidene fluoride, polyimide, and polytetrafluoroethylene [4,5,6,7].

PSF is considered one of the highest performing polymers because of its chemical stability, high mechanical strength, ease of modification, surface charge, and various operating temperatures and pH [8,9,10]. In the field of filtration, polysulfone membranes are used as ultrafiltration membranes to give high purification and as a pretreatment for reverse osmosis systems [11,12].

Furthermore, the hydrophobicity of PSF benefits selective absorption and organic component transportation. However, it is also noted as a major drawback, as the hydrophobic surface is responsible for attaching proteins and impurities, which further leads to membrane fouling [13,14]. Membrane modification by chemical and physical methods could be used to improve the hydrophilicity of the membranes. However, chemical modification may change the main chain of the polymer molecule, and the benefit of the membrane might be suppressed [15,16]). However, physical modification methods such as in situ blending using macromolecule materials are suggested for their suitability for mass production [17,18]. The introduction of nanoparticles during membrane formation is interesting because it offers a nanocomposite structure with improved mechanical properties, separation capabilities, and nanoparticle functionalities [19]. Much research related to the addition of nanoparticles such as Al_2_O_3_, TiO_2_, SiO_2_, and Ag into polymer membranes has indicated that the skin layer and support morphology could be affected by nanoparticles [20,21,22,23]. Silver nanoparticles had received the most consideration of these nanoparticles due to their promising features (hypoallergenic, antibacterial, and nontoxic) in manufacturing mixed matrix membranes [24,25,26]. Different methods have been used in fabricating membranes loaded with silver nanoparticles. Baldino et al. [27] incorporated Ag–Ha nanoparticles into poly(vinylalcohol) (PVA) membranes by supercritical Co2 (SC-CO2) phase inversion. The result showed that membranes fabricated at 20% w/w PVA showed a significant E. coli inhibition at a Ag concentration of 9 ppm. Han et al. [28] fabricated reduced graphene oxide (rGO) nanofiltration membranes by adopting a plasma assisted in situ photocatalytic reduction method. This method utilized Ag nanoparticles as a plasmonic photocatalyst to form rGO-based composite membranes. The result indicates the rGO–Ag membrane’s retention capacity, water flux, and stability are improved when treating toxic Cr(VI) solution. Mai et al. [29] prepared a Ag–PAN catalytic nanofiber composite membrane electrospinning process where Ag nanoparticles were immobilized on the nanofibers. The water permeability of the Ag–PAN analytic nanofiber composite membrane reached the highest value at 49.64% of the amount of Ag nanoparticle immobilization. The dry phase inversion method was also used to decorate silver nanoparticles onto polyhedral oligomeric silsesquioxane nanocages to improve the nanofiltration membrane structure and biological properties. The fabricated membranes exhibit an excellent water permeability and antibacterial properties [30].

Our extensive studies revealed that, regardless of the research that has worked on the effect of nAg size relative to the membrane, there is no considerable research that has sorted out the nanoparticle’s effect on the average pore size of the membrane, separation of low molecular weight protein (lysozyme; 12 kDa), and the ultrafiltration performance of the fabricated membrane. This work aims to study the effects of nAg size on the membrane pore size, morphology, protein separation, ultrafiltration performance, and antibacterial characteristics of the membrane. To achieve this, this work made a neat PSF membrane and Ag/PSF membrane using the phase inversion method. Ag/PSF membranes were prepared with 20, 40, and 90–210 nm of nAg particles in the casting solution. The effect of nAg particles on the pore statistics and hydrophilicity was investigated. The rejection of aqueous protein solutions, BSA, and lysozyme was also studied.

## 2. Materials and Methods

### 2.1. Materials

Polysulfone (PSF) for making ultrafiltration membrane was supplied from Solvay. Bovine serum albumin (BSA) (69 kDa) and lysozyme (12 kDa) were obtained from HiMedia Laboratories Pvt Ltd., Mumbai, Maharashtra 400086, India, coliform agar and NMethyl-2-pyrrolidone (NMP) was obtained from Merck KGaA, Darmstadt, Germany. Silver (20, 40, and 90–210 nm) nanoparticles, as shown in Figure 1 [31,32,33], were obtained from Nanostructured & Amorphous Materials, Inc., Katy, TX 77494, USA. Pure water was used for the membrane’s fabrication and water flux test.

### 2.2. Membrane Fabrication

The membrane was fabricated using the phase inversion approach. The procedure is as follows: nAg (0.22 wt.%) was dispersed into the solvent, NMP, to produce a dispersed nAg/NMP solution. Then, PSF was dissolved in nAg/NMP solution. The prepared solution of the PSF/nAg was casted on a glass plate using a film applicator (Elcometer, Manchester M43 6BU, UK) at a thickness of 200 microns. The glass plate and cast solution were gently transferred to a coagulation bath containing pure water at room temperature.

### 2.3. Membrane Characterization

The contact angle analysis was conducted by taking the water drop image using a Dino-lite digital microscope. The angle of the water drop was calculated using CAD software. The contact angle of each membrane was measured three times and the average values were collected. From the acquired contact angle values, the work adhesion $\omega_{A}$ (surface energy) required to drag water from a membrane surface (in square meters) can be calculated [34]:(1)$$\omega_{A}=\gamma_{B}(1+\cos\theta )$$
where $\gamma_{B}$ is the water surface tension ($7.2\times 10^{-2}\;N/m$) and θ is the contact angle.

### 2.4. Equilibrium Water Content

The fabricated membranes were cut into the requisite size, immersed in distilled water for 24 h, and weighed instantly after wiping free membrane surface water. The membranes were then dried out and weighed again. The water content of the membrane was determined by [35]: (2)$$\%\mathrm{WC}=\frac{W_{w}-W_{d}}{W_{w}}\times 100\;$$
where $W_{w}$ and $W_{d}$ are the weights of the wet and dry membrane, respectively.

### 2.5. Ultrafiltration Experiments

#### 2.5.1. Pure Water Flux Test

Ultrafiltration test was performed in stirred dead end cell (Sterlitech UHP-62, Ster-litech Corp., Kent, WA, USA), as illustrated in Figure 2. At a pressure of 2 bar, nitrogen gas was delivered into the test cell to pressurize the pure water to pass through the membrane pores. Data logger was used to record the weight of the permeated water coming through the membrane pores. The following equations were used to calculate the volumetric flux and permeability [36]:(3)$$Flux\;\left(J_{v}\right)=\frac{Q}{A\Delta t}$$
(4)$$Permeability\;\left(L_{p}\right)=\frac{J_{v}}{\Delta P}$$
where Q is the amount of the collected pure water (in *L*) during the sampling time, Δt (in h), A is the area of the membrane (in m^2^), and ΔP is the pressure difference (in bar).

#### 2.5.2. Protein Separation

In phosphate-buffered solution (pH = 7.2), 0.1 wt.% of BSA (69 kDa) and lysozyme (12 kDa) protein solution were prepared separately. The ultrafiltration experiment in protein rejection was conducted at a constant pressure of 2 bar. The concentration of protein (permeate) was measured using an N4S UV–visible spectrophotometer at a wavelength of 280 nm. The protein rejection (PR) was calculated by [37]:(5)$$\%\mathrm{PR}=\left[1-\frac{C_{p}}{C_{f}}\right]\times 100\;$$
where $C_{p}$ and $C_{f}$ are the concentration of proteins in permeate and feed solution, respectively.

#### 2.5.3. Measurement of Average Pore Size

The ultrafiltration experimental findings were used to determine the membrane average pore size at the surface. The average pore size membranes were measured using the molecular weight of the solute with a PR of more than 80% using the equation below:(6)$$\bar{R\;}=100\left[\frac{\propto }{\%SR}\right]$$
where $\bar{R\;}$ is the average pore size (radius) and ∝ is the solute radius, represented by the Stoke radius obtained from the solute molecular weight according to the slit sieve method [38].

### 2.6. Molecular Weight Cutoff (MWCO)

The molecular weight cutoff is stated to correlate linearly with the membrane’s pore size. In most cases, molecular weight cutoff of the membrane is examined by recognizing an inert solute of the lowest molecular that shows a PR of 80–100% in an ultrafiltration experiment. This experiment selected proteins of dissimilar molecular weight such as BSA and lysozyme for percentage rejection of PES/nAg blended membrane [39].

### 2.7. Antibacterial Experiment

To assess the effectiveness of antibacterial properties of the membrane, irrigation water was used as water sample to represent water contaminated by bacteria. The tested membranes were immersed in the contaminated water and then placed onto the EMB agar Petri dishes and incubated at 35 °C for 24 h. Agar was prepared by autoclaving the liquid media at 121 °C for 15 min and letting it cool down in sterilized Petri dish [40,41]. The numbers of bacterial colonies that appeared on the tested membrane were analyzed and reported.

## 3. Results and Discussion

### 3.1. Contact Angle Analysis

Figure 3 classifies the water contact angle of bare PSF, PSF/nAg-20, PSF/nAg-40, and PSF/nAg-90–210. As presented in Figure 3, more hydrophilic surfaces appeared after the introduction of the nAg particles. The highest contact angle (80.6°) was achieved for the bare PSF membrane, whereas the smallest nAg particles (20 nm) gave the lowest contact angle value (58.5°). The obtained contact angle values were then used to calculate the work of adhesion (surface energy), which was the same figure. The lowest value of surface energy (84.4 mN/m) was acquired for the bare PSF membrane, and the highest surface energy (110.7 mN/m) was obtained for PSF/nAg–20 nm, indicating that the introduction of the smallest nAg particles improved the membrane’s surface wettability [42,43]. This result also stated that the presence of nAg particles improves the hydrophilicity of the membrane surface. It was found that the contact angle increased with the introduction of nAg particles. Furthermore, it has been observed that the contact angle depends on the particle size in which the value increased as the size increased from 20 to 40 nm. However, for larger sized particles (40 to 90–210 nm), there may exist several factors such as porosity, surface roughness, and nanoparticle distribution, which can control the contact angle of the membrane’s surface [44].

### 3.2. Equilibrium Water Content Study

Figure 4 shows the equilibrium water content of the membranes. Through this figure, it can be concluded that there is no significant difference between any of the membranes. Normally, water content was affected by hydrophilicity; however, in this case, the presence of nAg particles altered the membrane’s hydrophilicity properties. The increase in hydrophilicity, however, does not significantly increase the membrane water content. This is because the membrane hydrophilicity was caused by the presence of nAg particles instead of membrane porosity [45,46].

### 3.3. Pure Water Flux Test Experiments

Figure 5 shows the pure water flux of various fabricated membranes. The pure water flux was decreased by introducing nAg particles into the membranes. The decrement was even higher for the PSF/nAg-90–210 membrane. This might be due to the pore blockage by the nanoparticles, as reported by Mollahosseini et al. [42], and a larger particle (90–210 nm) tends to create more blockage than the smaller one, resulting in the lowest pure water flux. Even though the three membranes have similar contact angle values, the membrane permeability among those three are comparatively different. It appeared that the pore blockage created by the larger particles had more influence on the pure water flux compared to the contact angle surface.

### 3.4. Protein Separation

Figure 6 shows the BSA rejection of the fabricated membranes. According to the trend describing the BSA rejection in this figure, three membranes (bare PSF, PSF/nAg-20, and PSF/nAg-40) give a similar value of BSA rejection, which lies in the range of 91–93%. This can be explained by the addition of nanoparticles (20–40 nm) significantly reducing the membrane’s pore size. However, the membrane embedded with the largest nAg particle size (PSF/nAg-90–210) had the lowest BSA rejection value (70.7%). This is due to the extra number of macromolecules passing through the membranes caused by the increment of the particle size.

Furthermore, the value of lysozyme rejection is below 75%, where the molecular weight of lysozyme (12 kDa) is smaller than BSA (65 kDa). The bare PSF and PSF/nAg-20 membranes gave similar values of lysozyme rejection of 69–70%, while PSF/nAg-40 and PSF/nAg-90–210 rejected the lysozyme for 48% and 27%, respectively. According to the lysozyme rejection result, the increment of particle size significantly decreased the membrane’s rejection. When the membrane was used to reject protein at a molecular weight of 65 kDa, the bare PSF, PSF/nAg-20, and PSF/nAg-40 membranes gave a similar value. However, when the molecular weight of the membrane was reduced to 12 kDa, the PSF/nAg-40 membranes could not give a value similar to the other two membranes (bare PSF and PSF/nAg-20). This is because the particle size of the membrane created a larger membrane pore size [47,48].

### 3.5. Measurement of Average Pore Size

Table 1 presents the summary of the average pore size. Based on the BSA rejection data, three membranes (bare PSF, PSF/nAg-20, and PSF/nAg-40) rejected the BSA protein up to 93%, which is high considering that those membranes have an average pore size of 38 Å. There is no significant difference in the average pore size by changing the particle size from 20 nm to 40 nm. However, for membrane PSF/nAg-90–210, the membrane failed to reject above 80% of the BSA solution in the water, which means than the PSF/nAg-90–210 has a pore size bigger than 38 Å, which is likely due to the utilization of bigger nAg particles.

Figure 7 shows the surface SEM micrographs of the bare PSF, PSF/nAg-20 nm, PSF/nAg-40 nm, and PSF/nAg-90–210 nm membranes. Relatively large pores were formed on the surface of the bare PSF membrane. Smaller pores were formed on the membrane surface when nAg particles with a size of 20 nm were introduced in the membrane dope solution. In addition, the PSF/nAg-40 nm and PSF/nAg-90–210 nm membranes have larger pores due to the use of a larger size of nanoparticles.

### 3.6. Molecular Weight Cutoff Measurement

The data for the molecular weight cutoff (MWCO) described in the experimental section are outlined in Table 1. As shown in Table 1, the MWCO values of the bare PSF, PSF/nAg-20, and PSF/nAg-40 membranes are 69 kDa. For the PSF/nAg-90–210 membrane, the MWCO value is higher than 69 kDa due to the performance of the membranes on BSA rejection being less than 80%. The molecular weight cutoffs of membranes tend to increase with an increased size of nanoparticles. A larger particle creates a larger hole on the membrane surface. Therefore, this larger hole will allow smaller particles to pass through the membrane pores, resulting in a low molecular weight cutoff.

### 3.7. Evaluation of the Antimicrobial Activity 

Irrigation water was used to test the antibacterial activity of different sizes of nanoparticles against bacteria. Irrigation water was chosen as a water sample because it is a source of bacterial contamination [49,50,51].

Figure 8 shows the antibacterial test results for the bare PSF and PSF/nAg blended membranes. Due to the rigidity of the membrane, some edges of the membranes could not properly attach to the agar surface. The reduction of bacteria in nAg membranes was quantitatively measured using the image processing software ImageJ developed by the National Institutes of Health [52]. The nAg membranes were tested and incubated for 24 h to detect bacteria growth. The acquired image of each membrane was binarized and carefully adjusted for the software to detect each growth. Prior image detection, the acquired image and the binarized image were compared to confirm detection validity.

It has been observed that the number of bacterial colonies identified as coliform (purple colonies) [53,54] on the bare PSF membrane was higher than that of PSF blended with nAg. Figure 8b–d clearly show that the number of bacterial colonies on the membrane surface containing nAg decreased compared to the bare PSF membranes. Based on the results from ImageJ software, compared to the bare PSF membrane, the quantitative reductions of bacterial colonies appearing on the membrane surface are 67%, 72%, and 63% for the PSF/nAg 20, PSF/nAg 40, and PSF/nAg membranes, respectively.

Additionally, the PSF/nAg membranes with sizes of 20 and 40 nm showed fewer colonies, indicating adequate antibacterial effects against coliform. However, the PSF/Ag 90–210 nm membrane showed a higher number of bacterial colonies on its surface. This could be explained by the fact that smaller nanoparticles lead to higher surface contact with bacteria due to the lower crystallinity as well as a higher surface-to-volume ratio. Ali et al. [12] and Tang et al. [55] also stated that the number of bacterial colonies decreased significantly when treated with Ag nanoparticles. This is because the interaction between silver and bacteria can modify the bacteria’s metabolic activity and finally lead to death [23,56,57].

## 4. Conclusions

The procedure for the fabrication of PSF/nAg blended membranes has been successfully demonstrated with different nAg particles. The contact angle measurement confirmed that PSF/nAg blended membranes were more hydrophilic than the bare PSF membrane. The addition of nAg particles to the membrane, pressurized at 2 bar, reduced the pure water flux because the particles blocked the membrane pores. The number of bacterial colonies on the PSF membranes decreased significantly compared to the bare PSF membrane, indicating that nAg particles had an excellent bactericidal effect. These results show that nAg particles might be conceivably used as a separating membrane agent.

## Figures and Tables

**Figure 1 nanomaterials-12-00388-f001:**
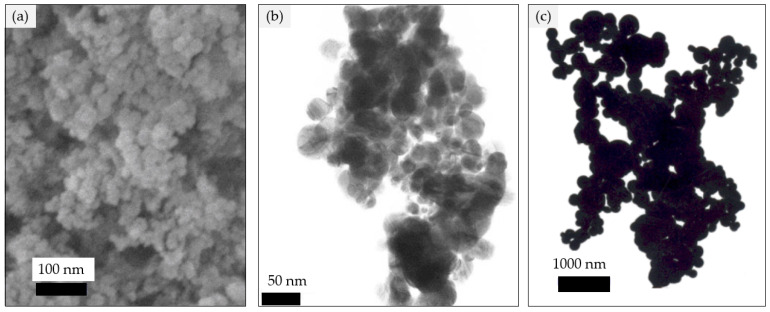
SEM image of silver nanoparticles (**a**) 20 nm, (**b**) 40 nm, (**c**) 90–210 nm.

**Figure 2 nanomaterials-12-00388-f002:**
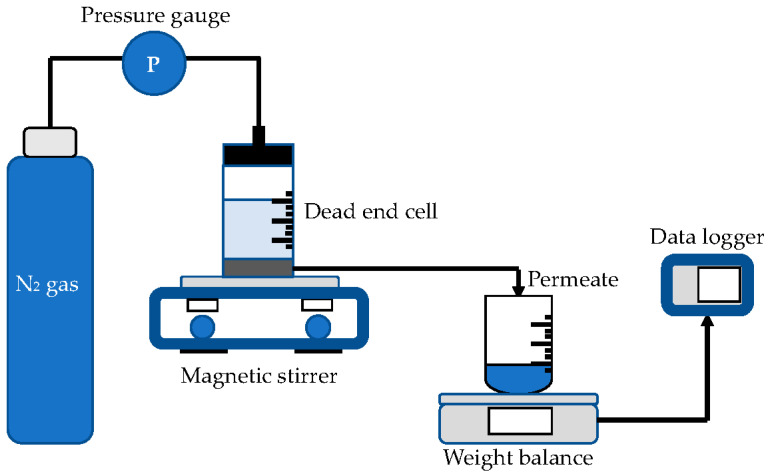
Experiment setup of pure water flux test.

**Figure 3 nanomaterials-12-00388-f003:**
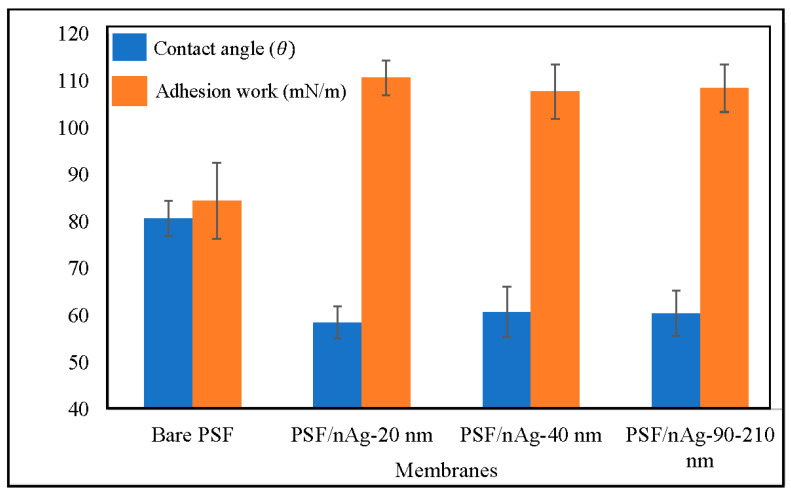
Contact angle and adhesion work of the bare PSF and PSF/nAg (20, 40, and 90–210 nm) blended membranes.

**Figure 4 nanomaterials-12-00388-f004:**
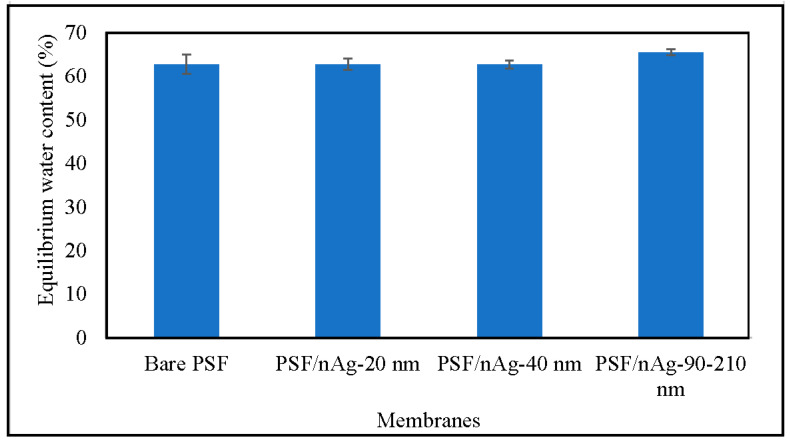
Equilibrium water content of the bare PSF and PSF/nAg (20, 40, and 90–210 nm) blended membranes.

**Figure 5 nanomaterials-12-00388-f005:**
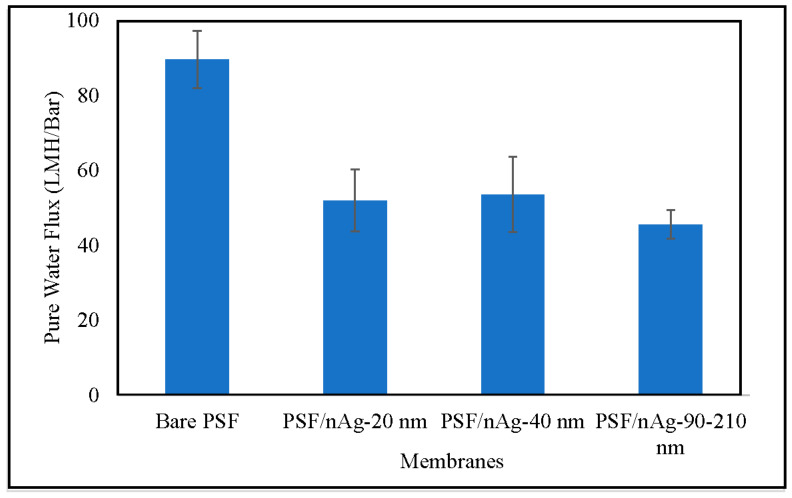
Pure water flux of the bare PSF and PSF/nAg (20, 40, and 90–210 nm) blended membranes.

**Figure 6 nanomaterials-12-00388-f006:**
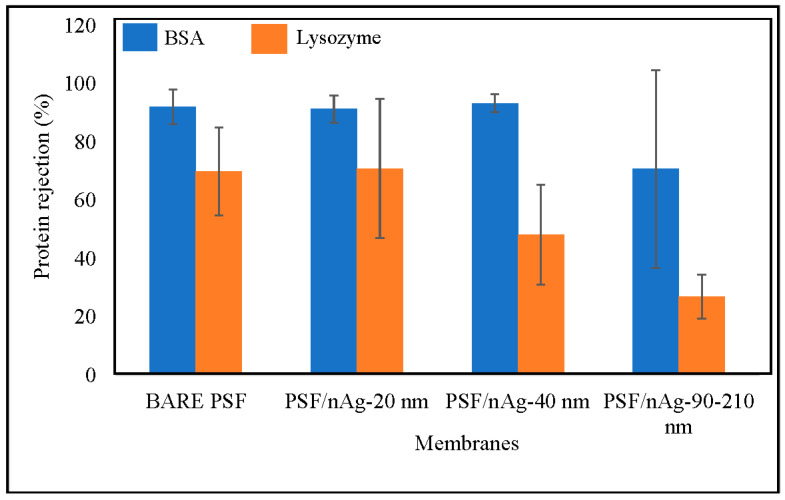
Effect of nanoparticles size on protein separation of the bare PSF and PSF/nAg (20, 40, and 90–210 nm) blended membranes.

**Figure 7 nanomaterials-12-00388-f007:**
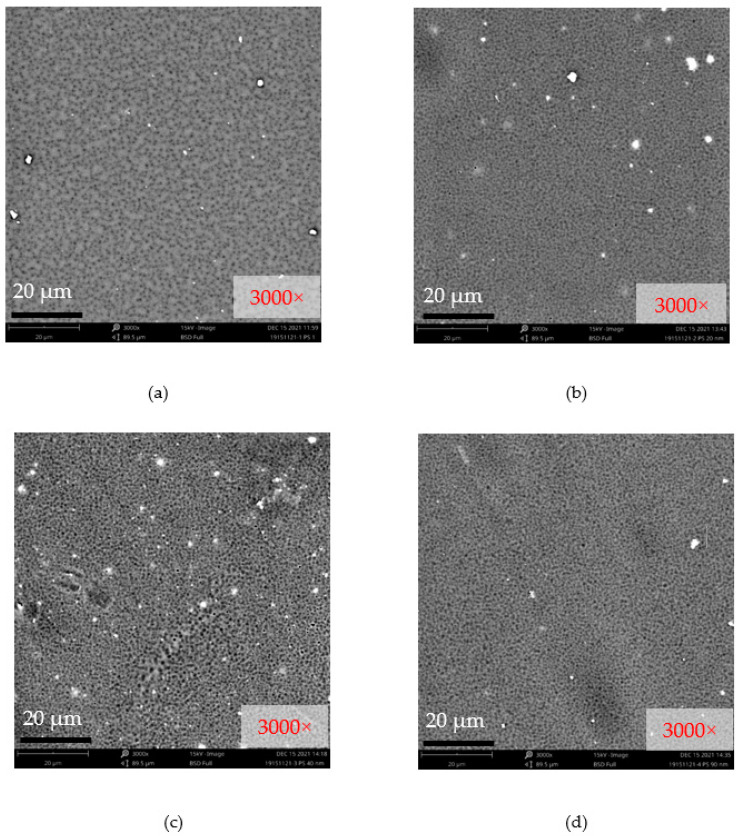
Surface SEM micrographs of (**a**) bare PSF, (**b**) PSF/nAg-20 nm, (**c**) PSF/nAg-40 nm, and (**d**) PSF/nAg-90–210 nm membranes.

**Figure 8 nanomaterials-12-00388-f008:**
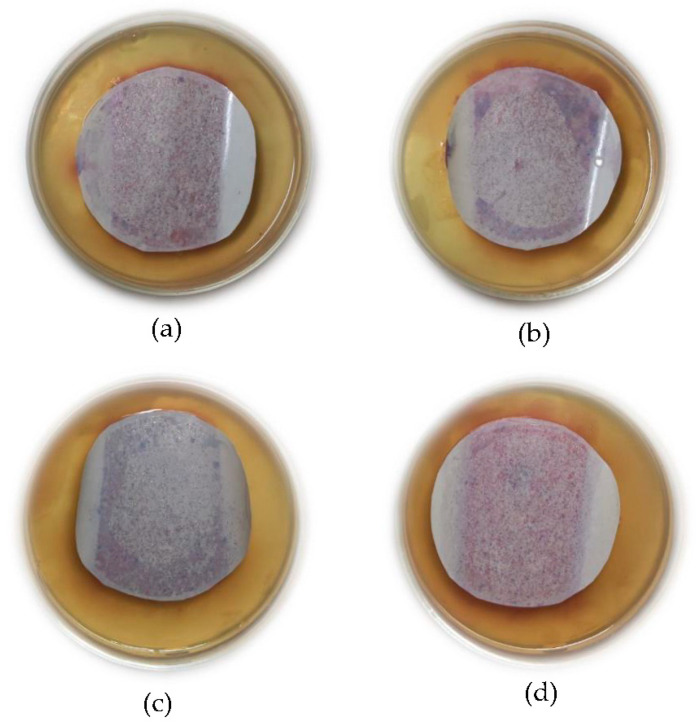
Result of the bacterial test, (**a**) bare PSF membrane, (**b**) PSF/nAg-20 nm, (**c**) PSF/nAg-40 nm, and (**d**) PSF/nAg-90–210 nm.

**Table 1 nanomaterials-12-00388-t001:** Pore statistics and molecular weight cutoff of PSF/nAg blended membranes.

Membrane Code	Pore Radius	MWCO (kDa)
Bare PSF	38.04	69
PSF/nAg-20 nm	38.5	69
PSF/nAg-40 nm	37.6	69
PSF/nAg-90–210 nm	NA	NA

## Data Availability

The data presented in this study are available on request from the corresponding author.

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
