# Peer review of "Study Effect of nAg Particle Size on the Properties and Antibacterial Characteristics of Polysulfone Membranes"

_nanomaterials, 2022, doi:10.3390/nano12030388_

Round 1

Reviewer 1 Report

This version is a lot better than the previous version. I have no further questions.

Author Response

Thank you for your comments and suggestions.

Reviewer 2 Report

I think that now the work is ok. Thank you

Author Response

Thank you for the comment and suggestion.

Reviewer 3 Report

The manuscript "Study effect of nAg particle size on the properties and antibacterial of polysulfone membrane" is accurate and well organized. Moreover, the topic is in line with the Journal scopes.

Some minor revisions are required, as follows:

- Correct typos (e.g., see 3th line in Abstract, etc...);

- M&M. If TEM images of Figure 1 are already published, this fact must be specified (e.g., ... as adapted from [...]);

- Error bars in Figure 3 and Figure 4 are not visible.

Author Response

This manuscript is a resubmission of an earlier submission. The following is a list of the peer review reports and author responses from that submission.

Round 1

Reviewer 1 Report

The authors tried to change the surface properties of the PSF film by adding silver nanoparticles. However, as the authors mentioned in the introduction section, such work has been reported. Moreover, the addition of silver nanoparticles causes the degradation of the filter membrane performance. I have not been able to see the unique features of this article, so I do not recommend the current version to be published. In addition, some suggestions are as follows:

(1) It would be better to give more convincing reasons to do this work.

(2) The fonts of the pictures in the article need to be unified.

(3) The testing of the experiment needs to be introduced in more detail.

Reviewer 2 Report

  1. There are many similar type of articles in the literature. The authors need to state the novelty of their work clearly.
  2. The introduction of nAg reduced the flux, didn’t improve the rejection, only improve the antibacterial properties. Please indicate the quantitative improvement (value or % improvement) of the nAg membrane in the abstract.
  3. In the keywords: what does the term “environmentally sound technologies” mean?
  4. Please indicate what type of images are present in Figure 1.
  5. In line 94: 2.4. Equilibrium Water Content: What is the significance of this study for a porous membrane?
  6. In section 2.7. Antibacterial Experiment: The antibacterial properties should be evaluated during UF experiment.

Reviewer 3 Report

line 29: please, modify abstract, because it seems like conclusions.

line 67: please, can you explain the % of 90-120nm silver particles? How are particles dimensions distributed and why 20 and 40 have no range of distribution?

line 99: please, specify Ww and Wd.

line 146: PSF/nAg-40nm and PSF/nAg-90-210 nm contact angles and adhesion work are about the same, how do you explain this result?

line 240: please, add the different behaviour of PSF/nAg membranes, related to nAg particles dimensions.

Reviewer 4 Report

The manuscript “Study effect of nAg particle size on the properties and antibacterial of polysulfone membrane” deals with the production of biopolymeric membranes by wet phase inversion, with antimicrobial properties thanks to the presence of silver nanoparticles. These membranes can be used for ultrafiltration. The idea behind the work is interesting and several analyses have been carried out on the produced loaded membranes; however, relevant improvements are required before the publication.

Detailed comments:

- Introduction. The state of the art related to the production of biocompatible and biodegradable membranes loaded with silver nanoparticles can be enlarged, adding other works that describe the use of alternative production techniques, such as supercritical phase inversion. For this purpose, see for instance this work: Baldino et al., Production, characterization and testing of antibacterial PVA membranes loaded with HA-Ag3PO4 nanoparticles, produced by SC-CO2 phase inversion, Journal of Chemical Technology and Biotechnology 2019, Volume 94, Issue 1, Pages 98 – 108; etc..

- R&D. Add a critical discussion to the results obtained since, in the current form, the work seems more similar to a report instead of a scientific paper. Moreover, a deeper comparison with the previous literature is required to underline the novelty and the relevance of the current findings.

SEM images of polysulfone membranes should be added to observe their morphology and porosity.